# Visibility Graph Analysis of the Seismic Activity of Three Areas of the Cocos Plate Mexican Subduction Where the Last Three Large Earthquakes (*M* > 7) Occurred in 2017 and 2022

**DOI:** 10.3390/e25050799

**Published:** 2023-05-15

**Authors:** Alejandro Ramírez-Rojas, Elsa Leticia Flores-Márquez, Carlos Alejandro Vargas

**Affiliations:** 1Departamento de Ciencias Básicas, Universidad Autónoma Metropolitana, Ciudad de Mexico 02128, Mexico; cvargas@azc.uam.mx; 2Instituto de Geofísica, Universidad Nacional Autónoma de Mexico, Ciudad Universitaria, Ciudad de Mexico 04510, Mexico

**Keywords:** visibility graph, seismic activity, subduction, tectonics, *k*–*M* slope

## Abstract

The understanding of the dynamical behavior of seismic phenomena is currently an open problem, mainly because seismic series can be considered to be produced by phenomena exhibiting dynamic phase transitions; that is, with some complexity. For this purpose, the Middle America Trench in central Mexico is considered a natural laboratory for examining subduction because of its heterogenous natural structure. In this study, the Visibility Graph method was applied to study the seismic activity of three regions within the Cocos plate: the Tehuantepec Isthmus, the Flat slab and Michoacan, each one with a different level of seismicity. The method maps time series into graphs, and it is possible to connect the topological properties of the graph with the dynamical features underlying the time series. The seismicity analyzed was monitored in the three areas studied between 2010 and 2022. At the Flat Slab and Tehuantepec Isthmus, two intense earthquakes occurred on 7 and 19 September 2017, respectively, and, on 19 September 2022, another one occurred at Michoacan. The aim of this study was to determine the dynamical features and the possible differences between the three areas by applying the following method. First, the time evolution of the *a*- and *b*-values in the Gutenberg–Richter law was analyzed, followed by the relationship between the seismic properties and topological features using the VG method, the *k*–*M* slope and the characterization of the temporal correlations from the *γ*-exponent of the power law distribution, *P*(*k*) ∼ *k*^−*γ*^, and its relationship with the Hurst parameter, which allowed us to identify the correlation and persistence of each zone.

## 1. Introduction

The dynamics of tectonic plates are considered to be one of the most important topics in geosciences, not only to improve scientific knowledge, but also to ensure the safety of small towns and big cities located in seismic zones considered to be high-risk. One of the main mechanisms of earthquake generation is convection in the interior of the Earth [1,2]. The convective processes involve nonlinear interactions which are far from equilibrium processes. Seismic processes originate from the interaction between plates in relative movement due to inner convection and strong correlations in time and space, as well as released energy or magnitudes ([3] and references therein). Although the vast majority of earthquakes occur in the interplate regions, the physical mechanism of the anthropogenic triggering of large earthquakes on active faults is connected with mining, artificial reservoir impoundment, geothermal operations, oil and gas field production and hydraulic fracturing (i.e., fracking) [4]. While almost all earthquakes occur at the plate boundaries, some anomalous events, called intraplate earthquakes, occur far from the boundaries [5]. The seismic mechanism can be considered to be a critical phenomenon exhibiting dynamic phase transitions [6], where a mainshock represents the new phase. In this context, seismic properties, such as the magnitude of an earthquake and the energy released during fracture processes, behave dynamically, showing short- and long-range temporal correlations, and these characteristics are governed by fractional Brownian processes and fractal/multifractal properties [3]. In fact, the complexity of the systems in the real world arises mainly from non-linear processes; thus, when the variability of a measurable parameter is registered in time, the system can be represented by a time series that can maintain some features of the underlying complex processes. In nonlinear signals analysis methods, there has been a trend of merging nonlinear time series analysis based on complex network theory.

In recent years, complex network theory has received important attention as a powerful tool to analyze complex time series [7,8,9,10,11,12,13,14,15,16,17,18,19]. In 2008, Lacasa et al. [16] introduced the Visibility Graph (VG) method, whose aim is to map a time series into a complex network or a graph. Under these transformations, it is expected that the topological properties of the network inherit the dynamical properties of the original time series and vice versa. The transformation of a time series into a graph entails building a geometric structure with two classes of sets, nodes and edges, where the nodes are the time series values and the edges determine the connectivity between nodes. The topological properties on a graph are defined by imposing conditions defining the connection between pairs of nodes by edges. In the case of the VG method, the connections between pairs of nodes are given by the reciprocal “visibility” between the values of the series [16], such that the connectivity between nodes can be defined. Visibility graph analysis has been applied as a powerful statistical method in extracting characteristic features of the time series, such as the periodicity, fractality, chaoticity and non-linearity [20]. For instance, some applications have been found in various research fields, such as medicine [21,22], economy and finances [23,24,25,26], seismology [27,28], oceanography [29] and meteorology [30,31], among others. Nowadays, the seismic activity in the world is continuously monitored by many seismic monitoring station networks, which have increased the amount of available seismicity data.

Among other methods, VG has allowed a deeper focus on the presence of dynamic features in the temporal distribution of seismic sequences [27]; a merit of the VG method is its ability to capture non-trivial correlations in non-stationary time series without introducing elaborate algorithms [7,32]. In addition, VGA has been shown to be a very useful tool to reveal the complex characteristics of seismic processes [20,27,28]. Because of its simple implementation, as well as its wide range of applicability, the VG algorithm has become popular in contrast with other methods. Telesca and Lovallo [20] were the first authors to apply the VG method to analyzing seismic sequences between 2005 and 2010 in Italy. These authors found that the VG method showed a collapse effect on the distributions of the degrees of connectivity of the seismic series with an increase in the magnitude threshold, which suggests that the properties of the distribution of the degree of connectivity can be independent of the magnitude of the threshold. In 2013, Telesca et al. [28] analyzed the seismic sequences monitored from 2005 to 2012 in five areas of the subduction zone in Mexico. In this investigation, an empirical link was found between the *b*-value of the Gutenberg–Richter law and the slope of the fitted line in the *k*–*M* plot by using the least-squares method. This outlined the relationship between *k* and *M* and was called the *k*–*M* slope. Other studies focused on the relationship between the *b* value and the *k*–*M* slope have been carried out, for example, in three dominant seismic areas of northern Iran [33], in seismogenic zones of Alaska, the Aleutian subduction zone [34] in Taiwan and Italy [35] and, recently, seismicity from Song Tranh 2 hydropower (Vietnam) was analyzed [20]. On the other hand, synthetic seismicity has also been generated experimentally in the laboratory through a system based on a stick–slip mechanical process, where two rough surfaces with different degrees of roughness move each other with different relative velocities [36], giving rise to avalanches that emulate earthquakes, and numerically by using the Olami–Feder–Christiansen numerical model [37]. In these cases, it has also been shown that the *b*-value is closely related to the *k*–*M* slope, suggesting that this relationship might be characterized by a kind of universality [20].

In addition, another dynamical property obtained from the VG method is the temporal correlation, which can be estimated by the *k*-degree distribution function. Kundu et al. [7] investigated the temporal correlations of two kinds of time series, a sequence of magnitudes and the inter-event times for three different types of seismicity: regular earthquakes, earthquake swarms and tectonic tremors. Their results were obtained by assessing the exponents of the *k*-degree distributions for the inter-event series, which showed a correlation similar to fractional Brownian motion. Additionally, they studied the time series of three different categories of earthquakes, from which they were able to distinguish topological features from the graph associated by using a visibility graph. As Telesca et al. [27] rightly point out, in all cited studies, the investigation of the relationship between the seismological and topological properties of seismicity has mainly been focused on tectonic or natural seismicity until now, because these regions still represent an open problem in the world.

The subduction areas located at Tehuantepec Isthmus, Chiapas (TCh), Flat Slab (FS) and Michoacan (Mi) belong to the Cocos plate, and each one has its own structural characteristics; their most important features are described in the next section. The Mexican subduction zone is a complex dynamical system with significant variation in slab geometry along the strike from northeast to southwest [38]. Some of these differences are the dip angle and subduction velocity, which are features that can determine the local seismic activity. The motivation for this work is because three large earthquakes occurred within these three zones recently: the first one (M8.2) struck on 7 September 2017, in the Tehuantepec Isthmus, which was considered an unusual event; the second one (M7.2) was a deadly event that occurred in the Flat Slab on 19 September 2017; and the third (M7.2) occurred on 19 September 2022, within the Michoacan zone. The coincidences in terms of month and day have drawn attention within the population, especially considering that two earthquakes, M8.2 and M7.9, occurred on 19 and 20 September 1985, respectively, causing huge losses of human life and damages in Mexico City. The aim of this paper is to study the seismic activity of the three zones by analyzing the catalogs from 2010 to 2022 using the VG method. First, the analysis is focused to identify the yearly variability of the *a*- and *b*-values, the seismic parameters of the Gutenberg–Richter Law, followed by the characterization of the three zones using the *k*–*M* slope’s behavior and, finally, the estimation of the temporal correlation from the *γ*-exponent associated with each *k*-degree distribution computed yearly and with the catalogs that comprise the whole period.

## 2. Tectonic Cocos Plate Settings

The Mexican subduction zone is a complex system characterized by variability in the shape of the subducted plate. Many authors have conducted significant studies into the crustal anisotropy and state of stress of the Mexican subduction zone ([39] and references therein). While in Jalisco, the Rivera plate subducts with a steep angle, in Michoacán, Guerrero and western Oaxaca, the Cocos plate becomes almost subhorizontal, changing again to a steep angle in the Tehuantepec Isthmus. This subducting activity also gave rise to the Trans-Mexican Volcanic Belt, an active continental volcanic arc that spans across Mexico, developed within an extensional tectonics setting. At the same time, there is new evidence of the geometry of the subducted slab being potentially due to subduction tearing or break-off. Therefore, the subducted slab is not laterally continuous but abruptly changes due to break-offs. Carciumaru et al. (2020) [39] presented a seismological study of the Earth’s crust using three different methods: azimuthal anisotropy based on ambient noise, shear-wave splitting of tectonic tremors and moment tensor inversions of the M8.2 earthquakes of 7 September 2017 in Tehuantepec, Mexico. In their work, they identify two slab tearings: in the Michoacan–Guerrero border and in Oaxaca, near the Tehuantepec fracture Transform/Ridge, the trench has an inflection point that slightly changes its trend to the southeast. At the Michoacan–Guerrero border, the subducted slab is subhorizontal, whereas in central Oaxaca the plate is characterized by northeast convergence. In this paper, we focus our seismic analysis on three regions of the subduction plate: Michoacan (Mi), Flat Slab (FS) and Tehuantepec Isthmus, Chiapas (TCh), because these regions better represent the slab geometry of the Mexican subduction zone and are examples of how the subducted plate presents variability in its shape and dynamics. These three regions are analyzed in terms of complexity measures in order to show their differences and to identify the characteristic values of the subduction regimes.

As was indicated by Carciumaru et al. in [39], the slab tearing at the Michoacan–Guerrero border constitutes an important change in the subducting slab. This tearing marks two different subduction regimes: in the north-west (Michoacan), where the subduction is perpendicular, the coast follows a constant slip and is less pronounced than in Jalisco, but not as flat as in Guerrero; and at the FS, inland at the southeast of the center of the Volcanic Belt, where the subduction shows a subhorizontal behavior. This flat slab area, referring to the subhorizontal shallow dipping lower plate, is delimited by considering the depths of the Mohorovicic interface between 40 and 60 km [40], a type of ellipse that circumscribes (or encloses) the earthquakes that occur in the area, and also where the intraslab earthquakes occur. Some authors [41,42,43,44,45,46] have already introduced the north Cocos tearing. However, the variability of the Cocos subducting plate has been well established by the bathymetric evidence of the Orozco and O’Gorman fractures and the Tehuantepec Transform/Ridge, and because the dip of the subducting plate varies from steep (in the Jalisco–Colima region) to flat (in central Mexico and at the southeast), there is a steep subduction in Chiapas (see Figure 8 in [39]). The Middle American Trench (MAT) also presents changes in velocity regimes and large dip variations along the strike [47]. Singh et al. [48] consider that this variability is precisely what explains the difference in the damage pattern in Mexico City, comparing the 2017 earthquakes and the disastrous interplate earthquake of 1985 (M8.0). The intraslab earthquakes occur closer to Mexico City, at greater depth, and involve higher stress drops than their interplate counterparts. Accordingly, the ground motion is relatively enriched at high frequencies as compared with that during interplate earthquakes, and damage is dominated by site effects [48].

The Michoacan region is characterized by high seismic activity, not only due to subduction events but also because of the existence of crustal faults in the interior. Of particular interest are the large earthquakes that have occurred in the region, such as the earthquakes on 19 and 20 September 1985, measuring M8.1 and M7.5, respectively, which occurred on a segment of the subduction zone known as the Michoacan gap [49]. This region is also responsible for intraplate earthquakes, such as those that occurred on 7 April 1845 and 19 June 1858 [50]. Moreover, the existence of crustal fault systems, such as those of Morelia–Acambay and la Paloma, have produced several earthquakes that have affected populations. The research carried out by [51] concludes that there are at least 316 seismically active faults in the region, derived from an extensional tectonics setting resulting from subduction. Additionally, the volcanic activity in the area is very important (as an example, the emergence of volcanoes: Jorullo in 1759 and Paricutín in 1943) and, as a consequence, the existence of associated seismic swarms.

After the Tehuantepec earthquake (M8.2), a particular interest arose in the Tehuantepec, Chiapas area. The sequence of earthquakes that followed after this earthquake occurred in two regions: one region is located at the Isthmus of Tehuantepec (coincident with the inferred location of the Tehuantepec Transform/Ridge [52] where slab tearing was reported [44,45,46]), and the other is considered as the aftershock zone (see Figure 1). According to [53], the Cocos plate convergence rate and directions at both sides (6.43 and 7.2 cm/year) of the Tehuantepec ridge are different. Mandujano-Velazquez et al. [54] and Keppie and Moran-Zenteno [55] investigated spreading rate changes, leading them to propose the existence of a microplate. This microplate would have been bound by the Tehuantepec Ridge and by a pseudo-transform fault.

## 3. Data Sets and Seismic Catalogs

The analyzed data sets, corresponding to the three seismic areas, were obtained from the National Seismic Service (SSN) website of the Universidad Nacional Autónoma de México (UNAM) (www.ssn.unam.mx (accessed on 10 November 2022) DOI:10.21766/SSNMX/EC/MX). The chosen periods for each monitored sequence were from 1 January 2012 to 10 November 2022. Within the selected areas, three large earthquakes occurred with *M* > 7.5: two in September 2017 and one in 2022. The first one, with an epicenter located within the Tehuantepec Gulf, struck on 7 September 2017 (M8.2) and was considered an unusual earthquake because it was in the intraplate. The second one, a deadly mainshock, had an epicenter in the Flat Slab area, a horizontal plate, located beneath the central area of Mexico and occurred on 19 September 2017 (M7.6) and the third and latest, whose epicenter was located in Michoacan State, struck on 19 September 2022 (M7.6). The areas where the three mainshocks occurred are indicated on the map shown in Figure 1. The catalogs were selected by considering the epicenters located in the areas for Mi, for FS and for the TCh.

The seismic activity of the three regions is shown in Figure 2, where the cumulative rate growth of the number of earthquakes monitored at Mi, FS and TCh can be observed. FS registered the minimum number of earthquakes at around 5000 over 13 years (Figure 2a in blue), whilst in Mi, around 15,000 earthquakes were registered (Figure 2a in red), and in TCh the number of earthquakes was approximately 75,000 (see Figure 2b) during the same period.

## 4. Methods

### 4.1. Gutenberg–Richter Law

The Gutenberg–Richter (GR) law describes an empirical relationship between the frequency and magnitude (*M*) of earthquakes in a specific region [56] following the distribution:(1)$$log_{10}\left(N\right)=a\;-\;bM$$
where *a* and *b* values are constants that characterize the seismic region, and *N* is the number of earthquakes with a magnitude ≥ *M*. It has been observed that both values, *a* and *b*, depend on the studied region and time. Changes in the *b*-value have been observed that are related to the spatial location of the analyzed area and to the time span observed. Additionally, changes in the *b*-value are inversely related to changes in stress [52]. In addition, in [57], a linear relationship between the *a*-value and *b*-value parameters of the Gutenberg–Richter (GR) law was shown, following *a~4b*.

### 4.2. Visibility Graph

This method considers a sequence of N events of a variable monitored evenly by a dynamical system. The VG means that any two events are connected by a right-line segment if they can see each other so that such a segment is not broken by any other intermediate value of the sequence. The VG maps each event as a node in a graph; therefore, each node connects with the other nodes based on the mutual visibility condition defined with the corresponding data heights. The visibility condition is expressed mathematically as follows: let two arbitrary data values (*ta*, *Ma*) and (*tb*, *Mb*), where *ta* < *tb*. Both points are visible to each other if any other data (*tc*, *Mc*) placed between them satisfy the visibility condition, as is shown in Figure 3, that is:(2)$$\frac{M_{b}-M_{c}}{t_{b}-t_{c}}<\frac{M_{b}-M_{a}}{t_{b}-t_{a}}$$
Lacasa et al. [16] showed that for this visibility condition, the associated graph of the time series is always (a) connected, (b) undirected and (c) invariant under affine transformations of the time series, including rescaling of both axes and horizontal and vertical translations. When the events represent earthquakes of magnitudes *M* such as seismic catalogs, the transformation is applied such that each earthquake of magnitude M_i_ represents a node in the graph, and the number of earthquakes that satisfy Equation (2) indicates the connectivity between earthquakes counting the *k*-degree (*k*-degree is defined as the number of direct connections between the *i*-vertex with the other ones; see Figure 3), and can also be associated with the temporal correlations of the data set in connection with the *k*-degree distribution of the nodes, as will be described in the next subsection.

### 4.3. k-Degree Distribution

In complex networks, the dynamical characteristics of the underlying system can be quantified by different measures; however, in the particular case of VG, one of the most interesting and important measures is the so-called vertex degree or *k*-degree: the number of direct connections between the *i*-vertex and the other ones. Because the graph is connected, each node has at least *k =* 1 degree for the first and the last nodes and *k* = 2 degrees for the other ones. In several works where VG has been applied to analyze time series, the connectivity has highlighted topological properties in the graph that are associated with the processes underlying the time series. This information is mainly given by the *k*-degree, as well as their distribution function, *P*(*k*). For different kinds of processes, *P*(*k*) displays specific behaviors; for periodic signals, VG is transformed as a concatenation of a finite number of motifs because the basic period is an integer multiple of the sampling rate. On the other hand, the opposite extreme case is white noise. For this process, the *k*-degree distribution behaves as an exponential distribution. For fractal time series, the *k*-degree distribution is generally scale-free following the power law:(3)$$P\left(k\right)\sim k^{-\gamma }$$
where the γ-exponent is related to the Hurst exponent *H* of the underlying time series as γ=3−2H for fractional Brownian motion and γ=5−2H for fractional Gaussian noise, so that it is possible to estimate a measure of the persistence and correlation, as well as the relationship with the β-spectral exponent from the power spectrum scaled as $f^{-\beta }$.

## 5. Results

### 5.1. Gutenberg–Richter Parameters

The local seismic activity is usually characterized by the Gutenberg–Richter law (Equation (1)), where the completeness magnitude “*M_c_*”, as well as the “*a*” and “*b*” parameters, are estimated. In this study, we employed the maximum likelihood method in the estimation of the *b*-value [27]. First, the *a*- and *b*-values for the entire catalogs of the three areas are listed in Table 1.

It has been shown that these parameters can change over time [58], showing possible fluctuations in the seismic activity of each zone. We analyzed the seismic activity of three different regions: Michoacan (MI), Flat Slab (FS) and the Tehuantepec Isthmus (TCh), as is shown in Figure 1. The period analyzed was from 2010 to 2022, in which the numbers of earthquakes registered were 14,003 in Mi, 5679 in the FS and 76,889 in TCh. Figure 3 indicates the growing rate of the occurrence of earthquakes. The temporal variability of the *b*-values calculated with the Gutenberg–Richter parameters for the activity of Mi, FS and TCh are shown in Figure 4. The *b*-value changed each year and displays an irregular pattern; nevertheless, in 2017, the *b*-values of FS and TCh attained low values, and after that, such values increased. Regarding the latest main shock at Mi in 2022, the respective *b*-value also decreased at a low.

.

The relationship between the *a*- and *b*-values of the Gutenberg–Richter law is depicted in Figure 5 and is in accordance with the result reported by [57]. It is important to point out that the expected relation *a~4b* is fulfilled in Mi and FS in accordance with [57]; however, for the TCh catalog, the relationship is given as *a = 4.52b*. This difference could be due to the post-seismic activity of the M8.2 main shock, which could be considered aftershocks and outside the tearing region of the Tehuantepec transform/ridge [39].

### 5.2. k-Degree vs. Magnitude Relationship

In order to characterize the topological and seismic properties between the graph and the seismic activity, Telesca et al. [36] introduced the *k*–*M* plane. The *k*–*M* slope is obtained as the mean square fitting of the right line in the *k*–*M* plane. The fact that the *k*–*M* slope is positive indicates a positive correlation; thus, the larger magnitude is then higher than the connectivity degree. The *k*–*M* slopes were calculated for the entire catalogs (including all magnitudes). In Figure 6a–c the *k*–*M* plane is shown for the catalogs of the three regions from the completeness magnitude threshold (*M* ≥ *Mc*). The differences in the *k*–*M* slope values are summarized in Table 2; in the third column, the reported values are from whole catalogs (*M* < *Mc*). The fitting in the *k*–*M* plot for the three regions can be considered linear when (*M* > *Mc*); however, for (*M* < *Mc*), the behavior is dominated by low seismicity and differs from linear fit, and is not reliable.

In Figure 6d, the variation of *k*–*M* slope is plotted versus the threshold magnitudes for the three catalogs. The first threshold magnitude, *M_th_*, is 3.2, and then increases by steps of 0.1; the maximum threshold magnitude depends on each catalog. In all cases, the *k*–*M* slope attains a maximum $M_{th}≅M_{C}$, after which the number of earthquakes becomes very small.

On the other hand, the *k*–*M* slope variability as a function of year is depicted in Figure 7 for the three catalogs. The *k*–*M* slopes values show an increasing trend for the three regions. FS and TCh stopped showing this trend in 2017, but Mi continued increasing until 2022.

### 5.3. P(k) Distribution

When estimating the correlation of seismic activity, this can be performed using the sequence of magnitude of earthquakes by applying the VG method to seismology, as has been reported in [20,34]. To do so, the seismic catalog is considered as a magnitude-discrete time series, rather than a continuous process, where the earthquakes are events with specified occurrence time. In fact, the catalogs are magnitude time series marked as temporal point processes that are described by the sum of Dirac’s delta centered on the occurrence time with amplitude proportional to the magnitude of the event; see Figure 3a, which shows a sequence of earthquakes in the catalog analyzed. In Figure 3b, the representation as a magnitude-discrete time series is displayed where the connectivity remains invariant. Next, the time series of magnitudes can be identified as a fractional Brownian motion so that the relationship between *γ* and *H* is valid [30]. The *k*-degree distribution (Equation (3)) of the visibility graph is assessed for the sequences of magnitudes of earthquakes monitored in the three regions. In [30], the authors showed the relationship between g-exponents in the *P*(*k*) power law, and the Hurst exponent was given by γ=3−2H. It is well known that the significance of *H* is determined by its deviations from the value of 0.5, which indicates randomness. The *k*-degree distribution *P*(*k*) vs. *k* in a log–log plane of each region was estimated for each one of the three regions, choosing the sequences as (a) the whole catalogs and (b) for the $M\;\geq \;M_{c},$ displayed in Figure 8a and b, respectively. The *P*(*k*) distributions display a behavior associated with correlated processes which are characterized by means of *H*. In Table 3, the γ-exponent values are summarized and in Figure 8a,b, the R^2^ adjust indicates the goodness of the γ-exponent fitted. It is observed that the γ-exponent of the Tehuantepec Isthmus is the largest, whilst for FS, it is the lowest in both cases.

### 5.4. Correlation Measure

Next, the *k*-degree distribution of the visibility graph was constructed (Equation (3)). The correlation of the seismic activity is directly related to the *γ*-exponent and *H*. A process is characterized as persistent if (0.5 < *H* < 1), antipersistent when (0 < *H* < 0.5) and uncorrelated or random if *H* = 0.5. In accordance with the relationship γ=3−2H, in Figure 9, the yearly variability of the *H*-values is displayed.

### 5.5. Three-Dimensional Plot: k–M Slope–b-Value–H-Exponent

In order to describe a generalized behavior, in this study we introduce a 3D plot (Figure 10a) in which the topological, seismic and dynamical properties of the three regions are identified.

The 3D plot, as shown in Figure 10a, condenses the information between the yearly variability and the whole catalogs with the $M\geq M_{c}$ of the three studied parameters: the topological properties represented by the *k*–*M* slope; the seismic features in terms of the *b*-value and the dynamical characteristic with the Hurst exponent (H). In addition, to represent the possible relationship between pairs of the three parameters, Figure 10b–d are the projections in three planes. Figure 10b, *k*–*M* slopes vs. *b*-value, shows a messy relationship between these parameters. Figure 10c, *k*–*M* slope vs. Hurst exponent, shows a poor linear fit in the sense of least squares with a negative slope. Finally, Hurst vs. *b*-value in Figure 10d also depicts some type of linearity with a negative slope. Regarding the parameters calculated for the entire catalogs with $M\geq M_{c}$ (stars) inside the purple ellipses, it can be seen that the yearly variability displays dispersion; however, a clear clusterization is observed when the entire catalogs are considered as marked by the purple ellipses.

## 6. Discussion

In this work, the study of three regions of the subduction Cocos plate was performed using the VG method. First, a study of the temporal behavior of the seismic parameters, the *a*- and *b*-values in the Gutenberg–Richter law, was performed. As has been reported in different works, both *a*- and *b*-values in the Gutenberg–Richter law change with time and region. When both seismic parameters are evaluated yearly, variability is found in both of them. The yearly variability of the *b*-value for the three regions (Mi, FS and TCh) is displayed in Figure 4, where it can be observed that FS and Mi attained the minimal value (*b* = 0.5) in 2011 and 2012, respectively, while in 2014 (*b* = 0.8), the lowest value was registered at TCh. This variation is mainly associated with the stress field as is discussed in [58 and references therein]. In Figure 4b, it can also be observed that the *b*-value decreased when the strong earthquakes occurred in 2017 at TCh and FS and also at Mi in 2022. In addition, in times when the number of seisms is low, the *b*-values seem to increase and remain so. Over the years, the behavior of the mean *b*-value is approximately 1.2 until 2016, but from 2017, this parameter increases. Concerning the relationship between *a* and *b*-values, Perez-Oregon et al. [57] obtained an analytical deduction of the positive correlation between parameters *a* and *b*, where $a=\left(4.01\;\pm \;0.02M\right)b+logC$ (where *M* is the magnitude and C is a constant), which is corroborated by the present analysis of the three zones. In Figure 5, the yearly relationship between both parameters for the three zones is shown; the relationship in [57] is fulfilled for FS and Mi, a≈4b, whilst for TCh, a ≈ 4.5b, it is slightly different than expected.

This difference in TCh is most likely due to the sequence of earthquakes following the M8.2 earthquake that occurred on 7 September 2017. The sequences occurred in two regions: one that is coincident with the inferred location of the Tehuantepec Transform/Ridge [52,54,55] where the slab tearing [44,45,46] is reported, and the other considered to be the aftershock zone [39].

On the other hand, the relationship between the topological properties and the seismic activity is identified for the three areas considered. As was introduced in [28], the *k*–*M* slope is the parameter that connects the topological properties and the seismic variability, which is calculated by the best linear fitting in the *k*–*M* plane, as is shown in Figure 6a–c. In this representation, the slope of the *k*–*M* relationship conveys information about the earthquake productivity of the seismic areas; thus, a measure of the correlation between the connectivity and magnitude is given by the *k*–*M* slope [28]. All the *k*–*M* plots show an increasing trend in the degree with the increase of the magnitude, as was shown by [36], which means that the *k*–*M* slope is a measure of the level of correlation. In addition, for the higher-magnitude events (*M* > 7), the relationship with the higher *k*-degree becomes almost one-to-one, as is shown in Figure 6a–c, where the *k*-degree is maximum. In the three studied cases, it can be observed that the behavior indicates that the correlations are positive. The numerical value of the *k*–*M* slope is a measure of the level of correlation and consequently of the degree of connectivity.

In Figure 6d, the *k*–*M* slope versus the threshold magnitudes is plotted for the three catalogs. The *k*–*M* slope attains maxima when $M_{th}\;≅\;M_{C}$, and afterward, the number of earthquakes becomes very small; therefore, it is not statistically representative.

Figure 7 shows the yearly behavior of the *k*–*M* slope, with $M\;\geq \;M_{c}$ for each case. It can be observed that the connectivity displays an increasing trend; nevertheless, FS and TCh stopped showing this trend in 2017, when the larger earthquakes occurred in September in each region. On the other hand, the connectivity of Mi kept increasing until 2022. This increasing r suggests that the connectivity could be associated with stress accumulation, and after the release, the variability of the connectivity is very low.

The distributions of *P*(*k*) vs. *k*-degree in logarithmic scale are shown in Figure 8a,b, where the γ-values are shown in the box inside for each catalog.

The *H* values are obtained from the γ-values. Figure 9 depicts the yearly *H* variability; from this behavior, it can be observed that all processes analyzed show short- and long-range temporal correlations, whose characteristics are governed by processes similar to fractional Brownian motion. The case of TCh activity shows antipersistence because *H* < 0.5 along the analyzed period. This behavior indicates that the dominant fluctuations come from low-magnitude seismicity over the 13 years studied. On the other hand, FS and Mi are persistent (*H* > 0.5); in the case of the seismicity of Mi, this can be observed from 2010 to 2014, and for FS, the persistence is between 2010 and 2017. After these periods, the behavior of Mi and FS changed to antipersistent processes from 2015 to 2022 and from 2018 to 2022, respectively. These behaviors suggest that the seismic activity in each region has different dynamics, whose conditions are established by the stress fields between the Cocos and North American plates, as illustrated in Figure 1 and has been reported in other studies [39,47,52,54,55].

Finally, a 3D plot (Figure 10a) is introduced with the aim to concentrate the information on the topological (*k*–*M* slope), seismic (*b*-value) and dynamical (*H*) properties of the three studied seismic sequences: FS, Mi and TCh. This plot contains the behavior of the yearly variability and, also, the value of the parameters, taking into account the entire catalogs with $M\geq M_{c}$. In this representation, we can perceive a dispersion of these properties from 2010 to 2022, but the same parameters estimated for the complete catalogs are located close to each other. When the 3D is projected, we obtain Figure 10b–d. Figure 10b conveys information about the earthquake productivity of the seismic areas. In terms of the dynamical properties of the sequences, the *k*–*M* slope vs. *b*-value displays a clusterization for *k*–*M* values above 14 and *b*-values ranging between 0.5 and 2.0; thus, it seems to be strongly related to the *b*-value of the Gutenberg–Richter law.

Figure 10c represents the projection of the *k*–*M* slope vs. *H*; it can be seen that TCh indicates antipersistence with high connectivity, while Mi also displays high connectivity while it is persistent or antipersistent and finally FS shows a mainly low level of connectivity when it is antipersistent or persistent but displays high connectivity only in some cases of persistence. Figure 10d displays the relationship between *H* and *b*-value; the main clusterization is located in the antipersistent area with *b*-values ranging between 0.5 and 2.0.

From these results, we can suggest that the VG properties of seismicity can resemble the seismological properties given by the parameters of the Gutenberg–Richter law. As already mentioned by [36], since the VG method takes into account the magnitudes of the events, this could suggest a way to analyze the statistical properties of seismicity more generally than the Gutenberg–Richter law.

As previously mentioned, Figure 10a, representing a 3D plot, was introduced with the aim of condensing the information between the yearly variability and the whole catalogs with $M\geq M_{c}$ of the three parameters studied: the topological properties represented by the *k*–*M* slope; the seismic features in terms of the *b*-value; and the dynamical characteristic of the Hurst exponent (H). The projections of data in three planes are also shown. When observing Figure 10c,d a poor linear fit can be seen in the sense of least squares with negative slopes. The parameters calculated for the whole catalogs with $M\geq M_{c}$ (stars) show a clear clusterization.

## 7. Conclusions

The aim of this study was to identify features and their possible differences between the three seismic areas of Mi, FS and TCh, belonging to the subducted Cocos plate and the North American plate. The selection of these zones was motivated because three large earthquakes (*M* > 7) have occurred in this area in the last five years: two of them on 7 and 19 September 2017 at TCh and FS, respectively, and the third in Mi on 19 September 2022. Our finding reveals the variability of the *b*-value in the GR law, as the *b*-value estimated yearly for each zone displayed large fluctuations, ranging from 0.5 to 2.5 in Mi and FS, and from 0.8 to 2 in TCh. It is worth noting that the *b*-value attains a minimum value in the years when the main shocks occurred, although it cannot be considered a generalization. The *a*-value in the GR law is related to the *b*-value as *a = 4b,* which is fulfilled for Mi and FS; nevertheless, for TCh, *a = 4.5b* was found, which indicates a difference in the seismic activity at Tehuantepec Isthmus. Regarding the seismic parameters, our findings suggest that the seismic activity at Tehauntepec Isthmus displays important differences with respect to Mi and FS. In terms of the topological and seismic properties, the numerical value of the *k*–*M* slope is a measure of the level of correlation and consequently of the degree of connectivity. TCh shows the lowest value in comparison with Mi and FS. The *k*–*M* slope values show an increasing trend for the three regions. FS and TCh stopped demonstrating this trend in 2017, after which their behavior was almost horizontal and parallel, but the Mi trend continued to increase until 2022. The *k*-degree distribution *P*(*k*) allows us to evaluate the temporal correlation of each sequence of magnitudes. TCh remains antipersistent during the period analyzed, while Mi and FS remain persistent, becoming antipersistent behavior, which suggests that the seismic activity of Mi and FS changed during the period analyzed. Finally, the 3D plot of *k*–*M* slope–*b*-value–*H* was introduced to represent the joint seismic, topological and dynamical properties of the three studied zones. The projections of data in the three planes of Figure 10c,d show a poor linear fit in the sense of least squares with negative slopes. The entire catalogs’ calculated parameters, for $M\;\geq \;M_{c}$ (stars), show clear clusterization.

## Figures and Tables

**Figure 1 entropy-25-00799-f001:**
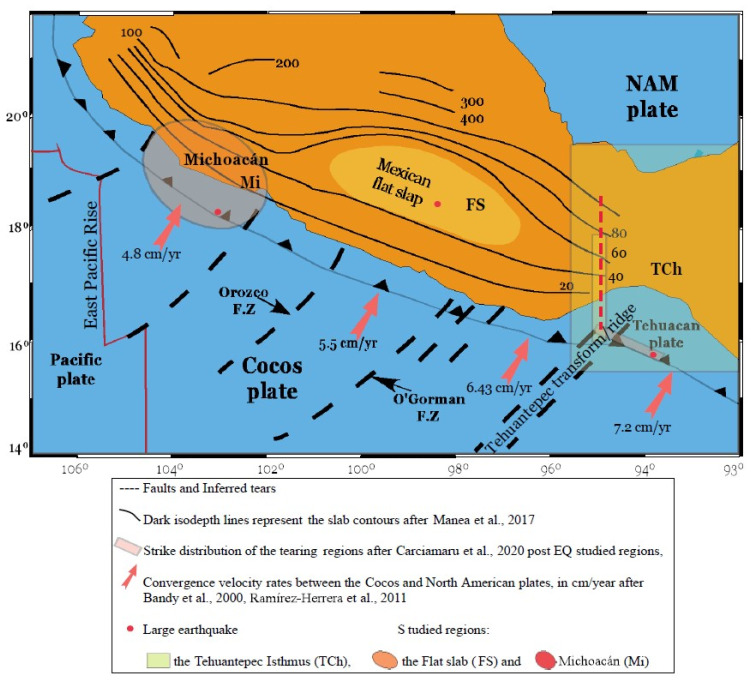
Mexican map. The three studied regions of the subduction Cocos Plate are depicted; also, the main tectonic features [39,40,41,47] and the larger earthquakes are shown.

**Figure 2 entropy-25-00799-f002:**
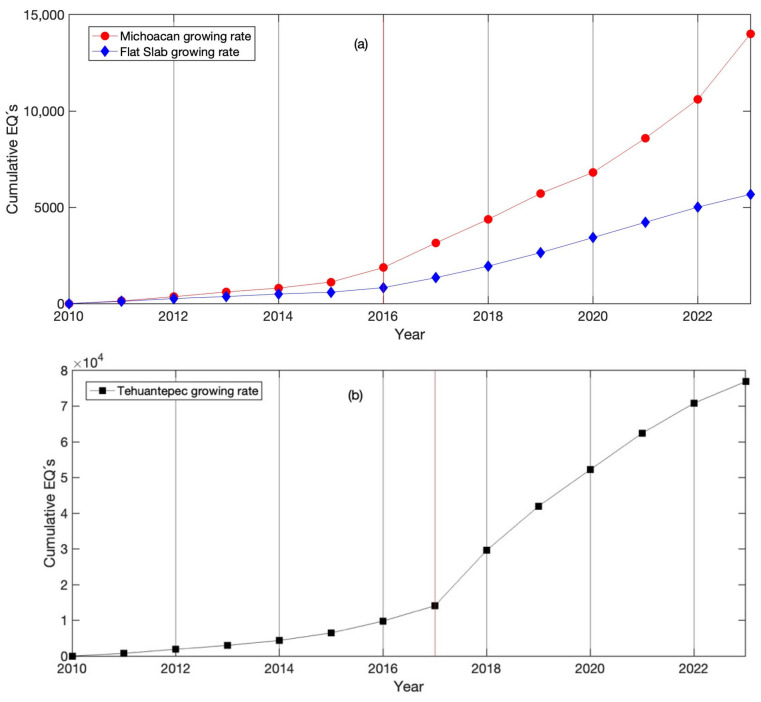
Cumulative number of earthquakes (EQs) per year; comparison between the three seismic regions. (**a**) The number of earthquakes in the Mi (red) and FS (blue) regions and (**b**) in the Tehuantepec Isthmus. The vertical line indicates the year when the activity increased: in 2016 at the Michoacan and Flat Slab and in 2017 at Tehuantepec.

**Figure 3 entropy-25-00799-f003:**
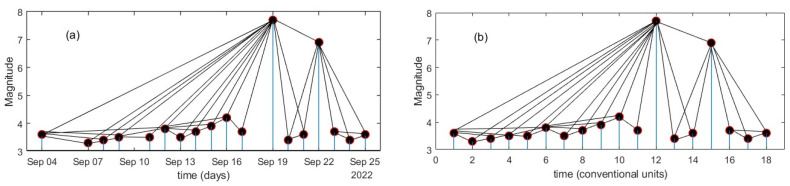
Example of the VG method of a seismic catalog. The black points are the magnitudes, and the black lines show the visibility condition. (**a**) VG for a seismic catalog as point processes; (**b**) the same catalog drawn as a discrete process where it can be seen that the connectivity does not change.

**Figure 4 entropy-25-00799-f004:**
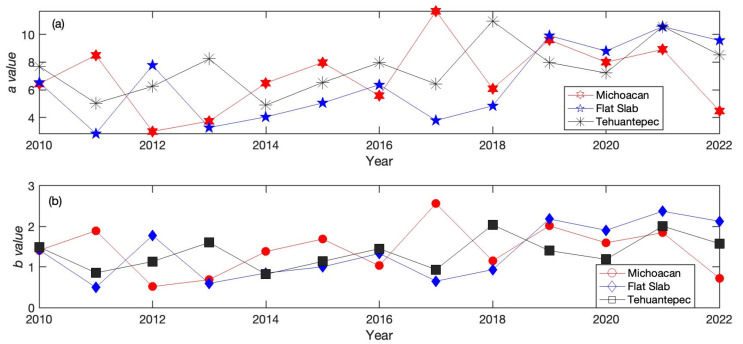
Yearly behavior of the seismic parameters (**a**) *a*-values, (**b**) *b*-values of the Gutenberg–Richter law, (**c**) completeness magnitude and (**d**) number of earthquakes with $M\;\geq \;M_{c}$.

**Figure 5 entropy-25-00799-f005:**
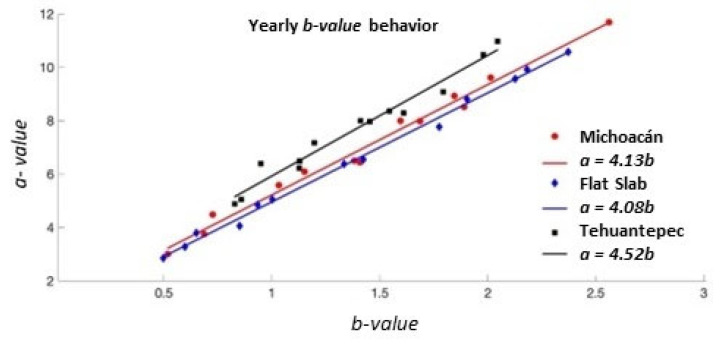
Relationship between the Gutenberg–Richter *a*-value vs. *b*-value.

**Figure 6 entropy-25-00799-f006:**
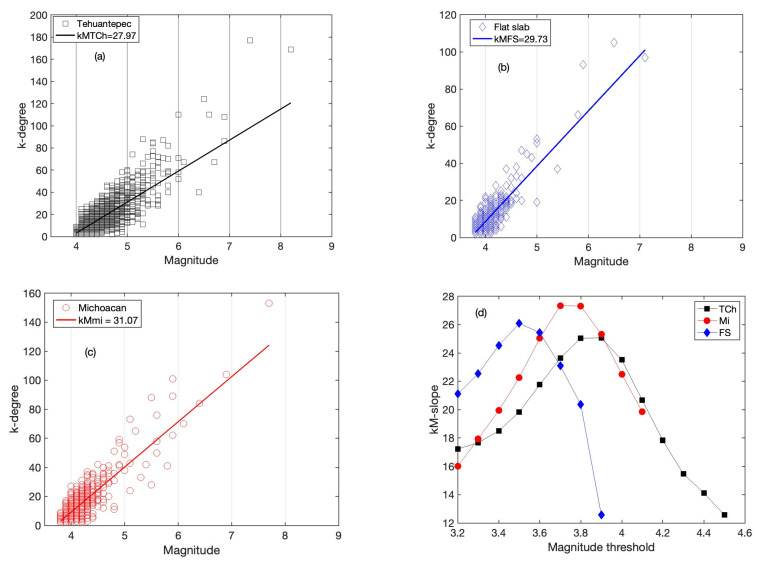
*k*–*M* plane of (**a**) TCh, (**b**) FS and (**c**) Mi. For the three cases, the calculation was estimated by using the completeness magnitude. (**d**) Variability of the *k*–*M* slope as a function of the magnitude threshold. The maximum values coincide with the completeness magnitude.

**Figure 7 entropy-25-00799-f007:**
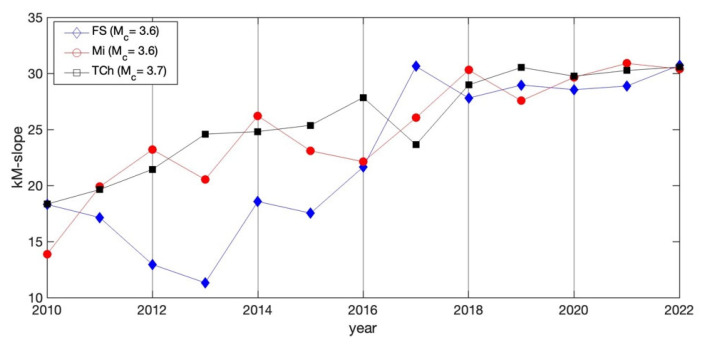
Yearly *k*–*M* slope variation of the three studied regions.

**Figure 8 entropy-25-00799-f008:**
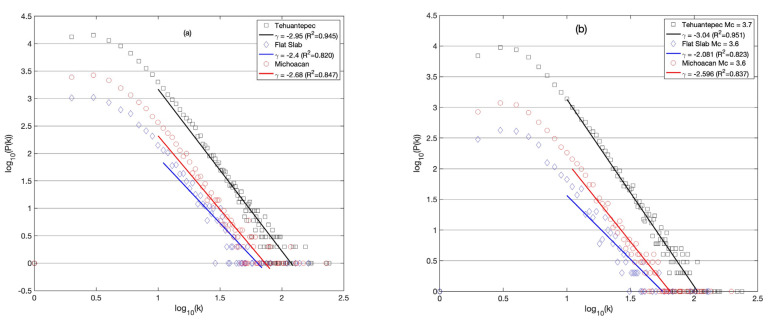
Distribution *P*(*k*) vs. *k*-degree in log_10_–log_10_ scale of the catalogs from 2010 to 2022. (**a**) Distributions of *P*(*k*) considering whole catalogs and (**b**) distributions of *P*(*k*) estimated with $M\geq M_{c}$ catalogs.

**Figure 9 entropy-25-00799-f009:**
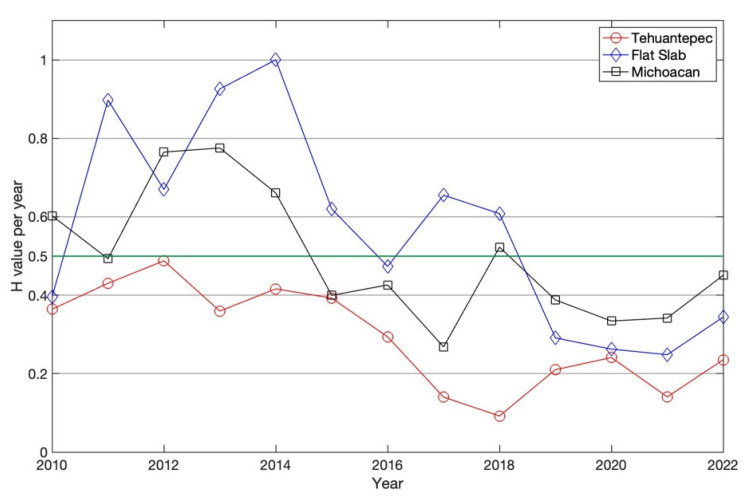
Hurst exponent behavior, obtained from relationship γ=3−2H for the three catalogs. The green line indicates the value *H* = 0.5, which indicates uncorrelation, which is the border between persistence and antipersistence.

**Figure 10 entropy-25-00799-f010:**
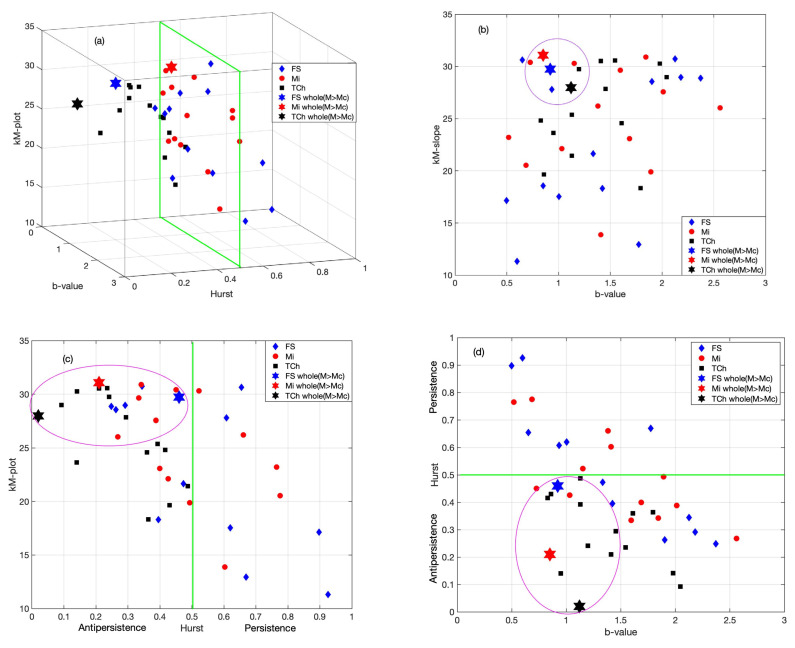
Each small point represents the yearly variation from 2010 to 2022 of the three seismic sequences: FS (blue), Mi (red), TCh (black) and the stars corresponding with the same parameters for the whole catalog with $M\geq M_{c}$. (**a**) Three-dimensional plot of topological (*k*–*M* slope), seismic (*b*-value) and dynamical (*H*) properties. (**b**) *b*-value versus *k*–*M* slope. (**c**) *H* versus *k*–*M* slope. (**d**) *H* versus *b*-value.

**Table 1 entropy-25-00799-t001:** Seismic parameters of the entire catalogs.

Region	*a*	*b*	*Mc*	*N* (*M* ≥ *Mc*)
Mi	5.43	0.85	3.7	4357
FS	5.53	0.92	3.5	3232
TCh	7.83	1.12	3.6	62,571

**Table 2 entropy-25-00799-t002:** Comparison of *k*–*M* slope between the entire catalog and with *M* ≥ *Mc*.

Region	*k*–*M* Slope with *M* ≥ *Mc*	*k*–*M* Slope Entire Catalog
Mi	31.07	15.94
FS	29.73	9.78
TCh	27.97	17.45

**Table 3 entropy-25-00799-t003:** Comparison of *γ* between the whole catalog and with *M* > *Mc*.

Region	*γ* Whole	*γ M* > *Mc* Catalog
Mi	2.88	2.58
FS	2.4	2.08
TCh	2.95	3.04

## Data Availability

Not applicable.

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
