# Peer review of "Visibility Graph Analysis of the Seismic Activity of Three Areas of the Cocos Plate Mexican Subduction Where the Last Three Large Earthquakes (M > 7) Occurred in 2017 and 2022"

_entropy, 2023, doi:10.3390/e25050799_

Round 1

Reviewer 1 Report

1) Before discussing the yearly variation of b-value (Fig. 4) it would be necessary to show the GR laws of the whole seismicity for each area.

2) A table would be useful where the completeness magnitude, the b and a value are reported for the whole seismicity for each area.

3) It is important to specify which method was used to estimate both the completeness magnitude and the b-value.

4) To complement the results shown in Fig. 4, it would be important to show also the yearly variation of a-value, Mc and number of events above Mc

5)I think it should be stressed that the relationship found by Lacasa et al. between VG-gamma (I mean the gamma of the p(k) obtained by VG)  and H is related to time-continuous processes and not to discrete processes, like those investigated in the present paper. So, if the authors aim to keep the section 4.4 and 4.5, they need to justify that the  VG-gamma-H relationship is also valid for discrete processes or, at least, specify that they adapting that relationship to the discrete case

6) Fig. 10 needs to be commented.

7) The sentence in the discussion (lines 455-456) "The values of the kM-slopes versus minimal magnitudes threshold, showed in Figure 7, display an increasing trend until 2017. After that, from the year 2018, the behavior becomes almost horizontal and parallel for the three regions. " would need further justification.

8) Line 460-471 discuss the persistence/antipersistence of the seismicity on the base of the relationship between H and VG-gamma. Again, it should be stressed that such conclusions could be stated just for time-continuous processes.

9) Line 464-465 " This behavior indicates that the dominant fluctuations come from low magnitude seismicity along the 13 years studied." Why this behavior is due to the low magnitude events?

10) The clusterization in Fig. 10 is not very clear. Actually, it seems that the clusterization is just governed by kMslope in (b) and (c): two clusters seem to be present: one given only by FS and the other given by MI-TCh. In (d) the clusterization seems not so much visible. 

The English must be improved, since many sentences are written in a contorted way rather than in a simple way. The lack of fluency makes the reading often not an easy task.

Reviewer 2 Report

The manuscript "Visibility Graph analysis of the seismic activity of three areas of the Cocos plate Mexican subduction where the last three large earthquakes (M > 7) occurred in 2017 and 2022" discusses the similarities and differences among three regions in the seismicity sense using the Gutenberg-Richter power-law, and network and visibility graph concepts. The introduction is well presented, although some basic concepts should be more adequately described. The manuscript can be better written as it determines some information that is unnecessarily repeated on several occasions. Please see some comments below.

(1) In Lines 35-37, the authors state, "One of the main mechanisms of earthquakes generation comes from the convection in the interior of the Earth.". Please include one or two references to support this information.

(2) In Lines 38-40, the authors state that "The seismic processes have their origin by the interaction between plates in relative movement, due to the inner convection, given place to strong correlations in time, space as well as released energy or magnitudes [1 and references therein]." The authors are referring to earthquakes at the boundary of tectonic plates. Although these earthquakes represent the vast majority, seismic sources should not be summarized as inter-plate earthquakes. In fact, in general, earthquakes happen due to the release of accumulated stress, as seen in human-induced (https://doi.org/10.1038/srep06100) and intraplate (https://doi.org/10.1785/0120160160) earthquakes. The authors should indicate these possibilities to avoid a particular concept about focal mechanisms.

(3) The first paragraph is too long. Consider breaking it into two paragraphs. The second can start, for example, in line 51("In the lately years, the complex network theory ...").

(4) Note that the information included in lines 57-58 ("where the nodes are the time series values and the edges determine the connectivity between nodes.") is the same as that contained in lines 60-61 ("each node represents a value of the series, and the connections between the nodes represent the relationships between them."). Consider writing them in a single sentence, avoiding unnecessary information repetition. The same happens at the beginning of section 3, where the information included in lines 244-245 ("It has been observed that both values, a and b, depend on the studied region and time.") is the same as that included in lines 245-246 ("Changes in the b-value have been observed that are related to the spatial location of the analyzed area and to the time span observed."). 

(5) In lines 111-112, "The subduction areas located at Tehuantepec Isthmus, Chiapas (TCh), Flat slab, central Mexico (FS) and Michoacan (Mi)," is the correct Flat slab (FS) and central Mexico (CS)? Furthermore, What is the correct? FlatSlab or FlatSlab? Also, TCh is Tehuantepec Gulf or Chiapas? The abbreviations are confusing. Please standardize the expressions in the new version of the manuscript. 

(6) What does the abbreviation EQ in the image and legend of Figure 1 mean? The abbreviations must be inserted in parentheses right after the written-out means when defined for the first time. 

(7) It is unclear what "cumulative rate growth of the number of earthquakes " means. The authors should clarify, as this concept is illustrated in Figure 2. Is it the time derivative of earthquakes' cumulative probability distribution (cdf) over time? The derivative with respect to the random variable of the cdf function? Please clarify this point.

(8) The Gutenberg-Richter law is actually Eq. (1) and not Eq. (2). Please double-check line 311.

(9) In Lines 317-318, the authors state, "The Figure 3 indicates the growing rate in the occurrence of earthquakes". Is Figure 3 not actually showing the VG method?

(10) In section 4.1, the authors discuss obtaining the parameters a and b of the GR law and the magnitude of completeness. Important information needs to be included in this selection. For example, what methods were used to estimate parameters a and b? How was the magnitude of completeness determined? Also, some plots showing cumulative distributions and the respective data fitting are appreciable.

Proofreading of the present manuscript is appreciated. 

Round 2

Reviewer 1 Report

accept

Reviewer 2 Report

The authors addressed discussed issues.